# Learning from What the Model Forgets: Prototype-Guided Patch-wise Replay for Medical Image Segmentation

## Abstract

Medical image segmentation remains a challenging problem due to the presence of hard positive samples that deviate from class centers and are frequently forgotten during training. These moderately forgettable samples often reside near decision boundaries and exhibit inconsistent learning behavior, contributing to elevated false negative rates and suboptimal boundary delineation. Existing methods lack effective mechanisms to identify and reinforce such samples, especially under patch-wise training constraints imposed by large-volume medical data. We propose an end-to-end online learning framework that systematically mines these moderately forgettable samples. Our method comprises three complementary modules: (1) Text-Guided Fusion, which incorporates CLIP-based text embeddings to guide semantic prototype learning and enhance feature representation; (2) Prototype-Based Scoring, which evaluates sample difficulty across intra-class consistency, inter-class distinction, prediction deviation, and model confidence; and (3) an Online Forgettable Sample Bank, which adaptively retains and replays informative samples through curriculum learning. Experiments on multiple public datasets demonstrate that our approach consistently reduces false negative rates and improves boundary accuracy in clinically challenging scenarios.

## 1 Introduction

Medical image segmentation is fundamental to computer-aided diagnosis. However, despite advancements in deep learning architectures Isensee et al. (2021); Hatamizadeh et al. (2022), models still struggle with complex anatomical structures near decision boundaries, particularly in ambiguous or low-contrast regions that are crucial for distinguishing anatomical boundaries Wang et al. (2019). Accurately identifying these challenging positive samples is critical for reducing false-negative rates in clinical practice Tang et al. (2024).

To understand sample difficulty, the concept of forgetting events was introduced Toneva et al. (2019), which tracks learning dynamics by monitoring when individual training examples transition from correct to incorrect classification during training, to categorize samples by their learning patterns Jagielski et al. (2022); Swayamdipta et al. (2020). Unforgettable samples are always correctly classified, while highly forgettable samples are frequently misclassified and typically correspond to noisy or extremely hard cases. Moderately forgettable samples, which repeatedly transition between being learned and forgotten (i.e., correctly and incorrectly classified) during training, represent cases of intermediate difficulty. These samples typically reside near decision boundaries and contain subtle yet informative features crucial for model generalization Mindermann et al. (2022); Benkert et al. (2022). Hard positive samples, which are essential for accurate boundary delineation, frequently fall into this category. Therefore, a principled approach to systematically identify and replay these moderately forgettable samples is essential for forcing the model to learn more robust representations, thereby enhancing segmentation accuracy Brignac et al. (2023).

Numerous hard sample mining approaches have been proposed, such as network modifications and loss optimizations Liu et al. (2024a); Li et al. (2023); Lin et al. (2017); Yeung et al. (2022); Salehi et al. (2017); Taghanaki et al. (2019). However, these often fail to address the intrinsic limitations of visual analysis for complex structures. As shown in Fig. 1, hard positive samples typically exhibit

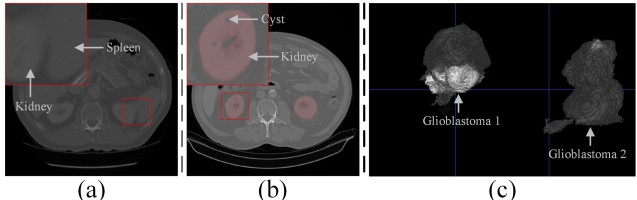

(a)        (b)        (c)

Figure 1: Examples of hard positive samples in medical images. (a) Low-contrast boundary between kidney and spleen in FLARE2021 dataset Ma et al. (2022). (b) Volume ratio between cyst (blue) and kidney (red) is approximately 1:100 in KiTS2023 dataset Heller et al. (2023). (c) Glioblastoma with high shape deviation in BraTS2020 dataset Menze et al. (2014), visualized from a coronal 3D perspective.

low-contrast boundaries, extreme size variation, or high shape deviation, making them difficult to distinguish. Multimodal semantic guidance provides a promising direction by incorporating textual knowledge to enhance visual understanding. In particular, pre-trained Contrastive Language-Image Pre-training (CLIP) Radford et al. (2021) aligns images and texts within a shared semantic space, facilitating effective representation learning. By leveraging CLIP text embeddings, models can capture subtle semantic distinctions often missed by visual features alone, demonstrating effectiveness in medical imaging applications and improving recognition of challenging cases Liu et al. (2023); Wang et al. (2022); Zhao et al. (2024).

Additionally, existing strategies developed for natural images Bengio et al. (2009); Kumar et al. (2010); Fan et al. (2017) often encounter computational bottlenecks when applied to three-dimensional medical volumes Zhu et al. (2019). To alleviate this, patch-based mining techniques He et al. (2021); Chen et al. (2024a); Isensee et al. (2021) divide large medical volumes into smaller patches for localized training. However, such local training overlooks global anatomical structures, limiting contextual consistency across patients and reducing sensitivity to critical boundary regions. Prototype learning has emerged as a promising alternative, enabling the learning of class-specific representative features Liu et al. (2024c); Zhu et al. (2024). However, existing works have yet to investigate its capacity to identify moderately difficult samples, which are also critical for improving model robustness.

Therefore, we propose an end-to-end online learning framework that systematically identifies and reinforces moderately forgettable samples during training. Our main contributions are as follows:

- We introduce **Text-Guided Fusion**, which leverages frozen CLIP text embeddings to guide visual-semantic prototype learning. This approach facilitates the generation of representative class centers, enabling improved identification of challenging positive samples near class decision boundaries.

- We develop **Prototype-Based Scoring**, which evaluates sample difficulty using four metrics: intra-class consistency, inter-class distinction, prediction deviation, and prediction confidence. These semantically-enhanced class prototypes robustly identify moderately forgettable samples.

- We propose an **Online Forgettable Sample Bank**, which dynamically maintains and replays informative samples through curriculum learning principles. This mechanism enhances model attention to critical features and mitigates repeated forgetting.

## 2  RELATED WORK

This section reviews methods for hard example mining. We first survey established approaches, categorized into loss- and model-based methods and sampling-based strategies. We then discuss recent advances in vision-language and foundation models to situate our work within the broader landscape of medical image segmentation.

## 2.1 Loss Function and Model-Based Methods

This category includes loss modifications and architectural improvements to emphasize hard positive samples. Online Hard Example Mining (OHEM) Shrivastava et al. (2016) and Focal Loss Lin et al. (2017) prioritize samples with high training loss, while SegLossBias Liu et al. (2024a) and Region-related Focal Loss (RFL) Li et al. (2023) leverage anatomical priors and region size. Dice-based losses Fidon et al. (2018); Salehi et al. (2017), Combo Loss Taghanaki et al. (2019), and Unified Focal Loss Yeung et al. (2022) address class imbalance and boundary sensitivity through adaptive weighting. Metric learning frameworks, such as triplet and contrastive losses Wu et al. (2017); Simo-Serra et al. (2015), further enhance sample separability. Architectural modifications, including MDNet-Vb Chen et al. (2021) and I2I-3D Merkow et al. (2016), improve fine structure perception but increase computational complexity. Despite these advances, existing methods often rely on short-term feedback, struggle with dynamic anatomical variation, and lack scalability for complex cases.

## 2.2 Sampling-Based Methods

Sampling-based methods aim to efficiently select informative or hard samples. Random sampling Wang et al. (2021) is prevalent but suboptimal. Active strategies Liu et al. (2021); Sun et al. (2022), Monte Carlo tree search Canévet & Fleuret (2016), dual-branch filtering Cho et al. (2019), and block-cyclic decomposition Henriques et al. (2013) improve selection but introduce computational overhead and instability. Curriculum learning Bengio et al. (2009); Fan et al. (2017) and importance sampling Katharopoulos & Fleuret (2018); Richtárik & Takáč (2016) adapt sample difficulty over time but offer limited representation in high-dimensional medical data. Patch-based online mining He et al. (2021); Chen et al. (2024a) integrates shape priors and bandit algorithms but is constrained by static templates and local context. In summary, sampling strategies lack global semantic understanding and struggle with identifying samples near decision boundaries, motivating our prototype-aware scoring and text-guided fusion framework.

## 2.3 Vision-Language and Foundation Models

**VLMs in Medical Imaging.** Large-scale pre-trained VLMs, particularly CLIP Radford et al. (2021), have been adapted for medical tasks to leverage their rich semantic understanding. Prior works adapt VLMs for medical imaging, using text prompts as inference-time queries for zero-shot or referring segmentation Chen et al. (2024b); Liu et al. (2023); Wu et al. (2023); Zeng et al. (2024). In these methods, the primary objective is to solve a direct visual-textual alignment task for a given image. In contrast, we employ language as a training-time semantic guidance. Our goal is not to segment based on a text query, but to inject semantic priors into the feature space itself, creating robust class-level visual prototypes. This process constructs a semantically structured feature space where visually diverse instances of the same class are compactly clustered. A well-structured space is a prerequisite for our core contribution: reliably identifying hard-positive samples based on their feature-space distance to these semantic anchors.

**Foundation Models.** While foundation models like the Segment Anything Model (SAM) Kirillov et al. (2023) and its medical variants (e.g., MedSAM Ma et al. (2024), SAM-Med3D Wang et al. (2024a)) excel at class-agnostic, promptable segmentation, they operate as powerful interactive tools. They address the challenge of delineating an object specified by a user. Our work, however, targets fully automated semantic segmentation, a non-interactive task where the model must learn and differentiate intrinsic anatomical semantics across a cohort. The inspiration from foundation models lies in their use of a powerful internal feature representation. Similarly, our work focuses on learning a robust feature space, but one that is optimized for automated semantic differentiation and the identification of hard samples, a critical need for advancing automated diagnostic pipelines where user-in-the-loop interaction is not feasible.

## 3 Method

We propose an end-to-end framework for medical image segmentation that explicitly mines moderately forgettable hard positive samples (see Fig. 2). The framework integrates three synergistic mod-

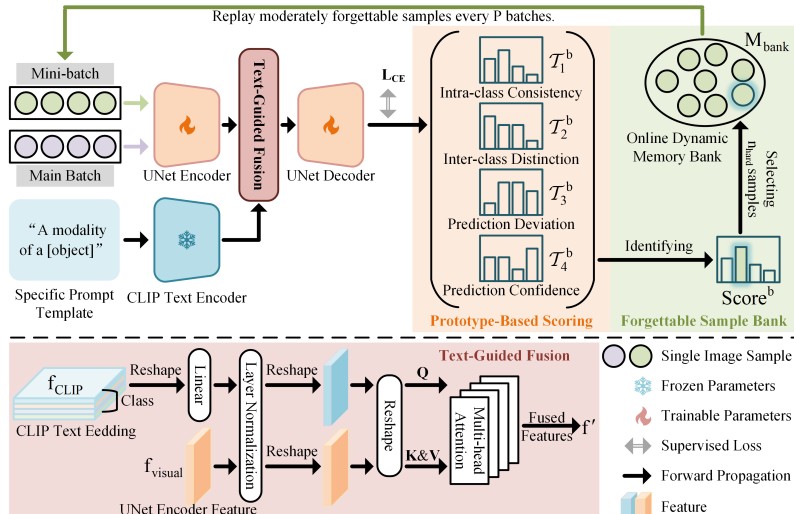

Figure 2: Overview of our proposed framework, which integrates a UNet encoder-decoder architecture with Text-Guided Fusion, Prototype-Based Scoring, and a Forgettable Sample Bank.

ules: (1) **Text-Guided Fusion** employs frozen CLIP text embeddings to enhance visual-semantic prototype learning that facilitates subsequent generation of representative class centers for improved challenging sample identification; (2) **Prototype-Based Scoring** conducts multi-dimensional sample difficulty assessment through four metrics with semantically-enhanced prototypes, thereby identifying informative samples that balance learning difficulty and informativeness; (3) an **Online Forgettable Sample Bank** retains and replays informative samples with semantic enhancement to reinforce attention to critical features and mitigate repeated forgetting. The following sections detail each component.

## 3.1 TEXT-GUIDED FUSION

Hard positive samples in medical image segmentation commonly emerge from complex anatomical structures, including lesions and organs that exhibit high intra-class variability, ambiguous boundaries, or morphologically similar yet semantically distinct regions, as illustrated in Fig. 1 and Fig. 2. These challenging cases often reside near decision boundaries where visual features alone provide insufficient discriminative information. Given that vision-only models struggle to capture the rich semantic relationships essential for robust medical segmentation Liu et al. (2023), we leverage the CLIP text encoder Radford et al. (2021) to incorporate external semantic knowledge. This enhancement is crucial for forming semantically robust visual features, which serve as the foundation for the class prototypes used in our subsequent difficulty scoring.

For each foreground class, text embeddings are generated using imaging-modality-specific prompt templates, such as "A magnetic resonance imaging of a [object]" or "A computerized tomography of a [object]", inspired by biomedical language models like BioLinkBERT Yasunaga et al. (2022). These anatomical-focused prompts leverage CLIP's pre-trained knowledge of normal anatomical structures, providing stable semantic cues that generalize across different pathological variations within each class. The resulting embeddings, $\mathbf{f}_{\text{CLIP}} \in \mathbb{R}^{(C-1) \times L}$, where $C$ is the total number of classes and $L$ is the embedding dimension, represent the text features for all foreground classes (excluding background). The use of such generic and anatomy-focused prompts is motivated by their ability to elicit more representative and robust class features from CLIP, reducing bias from dataset-specific terminology and improving generalization across diverse clinical scenarios. These text features guide prototype learning, which constructs representative features for each semantic region and improves segmentation discrimination Liu et al. (2024b); He (2024).

These embeddings then undergo a linear transformation and layer normalization (LN):

$$\mathbf{f}_{\text{CLIP}}^{\text{new}} = \text{Reshape}(\text{LN}(\text{Linear}(\text{Reshape}(\mathbf{f}_{\text{CLIP}})))) \tag{1}$$

Visual features $\mathbf{f}_{\text{visual}} \in \mathbb{R}^{B \times d \times H' \times W'}$, where $B$ is the batch size, $d$ is the feature channel dimension, and $H', W'$ denotes the spatial resolution after encoding, are processed similarly:

$$\mathbf{Q} = \text{Reshape}(\mathbf{f}_{\text{CLIP}}^{\text{new}}) \in \mathbb{R}^{(H' \times W') \times B \times d} \tag{2}$$

$$\mathbf{K}, \mathbf{V} = \text{Reshape}(\text{LN}(\mathbf{f}_{\text{visual}})) \in \mathbb{R}^{(H' \times W') \times B \times d} \tag{3}$$

The transformed text features are then spatially broadcast to match the visual feature dimensions, forming the query tensor $\mathbf{Q}$. Here, $\mathbf{Q}$ denotes the query (text-derived), while $\mathbf{K}$ and $\mathbf{V}$ are the key and value matrices (visual-derived), all reshaped for multi-head attention. We adopt multi-head attention Vaswani et al. (2017) to enable the model to jointly attend to information from multiple representation subspaces and capture complex cross-modal interactions between semantic and visual cues:

$$\mathbf{f}' = \text{Reshape}(\text{MultiHead}(\mathbf{Q}, \mathbf{K}, \mathbf{V})) \in \mathbb{R}^{B \times d \times H' \times W'} \tag{4}$$

This semantic fusion enriches visual features with anatomical knowledge. These semantically-informed features are then propagated through the UNet decoder, ensuring that the final output features, $\mathbf{f}_{\text{output}}$, inherit this enhanced context. The resulting prototypes, computed from $\mathbf{f}_{\text{output}}$, therefore capture both visual and semantic information, improving hard positive identification near class decision boundaries and providing a more stable basis for difficulty assessment.

## 3.2 PROTOTYPE-BASED SCORING

Building on semantically-enhanced features from Text-Guided Fusion, we develop a Prototype-Based Scoring mechanism to identify moderately forgettable samples. Traditional gradient- or loss-based approaches often rely on short-term feedback and suffer from computational overhead Shrivastava et al. (2016). While patch-based methods He et al. (2021) provide computational efficiency for medical images, they lack global semantic context for accurate difficulty assessment. Our approach addresses these limitations by leveraging semantically-enhanced prototypes to provide both computational efficiency and semantic-aware patch-level scoring.

Specifically, the UNet decoder output $\mathbf{f}_{\text{output}} \in \mathbb{R}^{B \times C \times H \times W}$ and ground truth $\mathbf{y} \in \{0, 1, ..., C-1\}^{B \times H \times W}$ are used to construct class prototypes:

$$\boldsymbol{\mu}_c^{\text{current}} = \frac{\sum \mathbf{f}_{\text{output}} \cdot \mathbf{m}_c}{\sum \mathbf{m}_c} \tag{5}$$

where $\mathbf{m}_{b,c,h,w} = \mathbb{I}[\mathbf{y}_{b,h,w} = c]$ is the class mask and $\mathbb{I}[\cdot]$ denotes the indicator function.

To alleviate the instability of prototypes caused by noisy predictions in individual mini-batches and to better capture temporal dynamics during optimization, we update prototypes using an Exponential Moving Average (EMA):

$$\boldsymbol{\mu}_c \leftarrow \beta \boldsymbol{\mu}_c + (1 - \beta) \boldsymbol{\mu}_c^{\text{current}} \tag{6}$$

We set the EMA coefficient $\beta = 0.99$ to ensure stable yet responsive adaptation of prototypes during training He et al. (2020). Masked features $\mathbf{f}$ and masked probabilities $\mathbf{p}$ are defined as $\mathbf{f} = \mathbf{f}_{\text{output}} \cdot \mathbf{m}$ and $\mathbf{p} = \text{softmax}(\mathbf{f}_{\text{output}}) \cdot \mathbf{m}$, respectively. This masking ensures that the subsequent scoring metrics focus exclusively on pixels belonging to their ground-truth class, linking the sample's score directly to its positive-class representation.

To comprehensively capture sample difficulty near class decision boundaries, we evaluate samples across four normalized dimensions: (1) intra-class consistency ($\mathcal{T}_1^b$), (2) inter-class distinction ($\mathcal{T}_2^b$), (3) prediction deviation ($\mathcal{T}_3^b$), and (4) prediction confidence ($\mathcal{T}_4^b$). $\mathcal{T}_1^b$ measures the dispersion between pixel features and their class prototype; $\mathcal{T}_2^b$ quantifies proximity to other class prototypes; $\mathcal{T}_3^b$ assesses mismatch between prediction and ground truth; $\mathcal{T}_4^b$ assesses the model's lack of confidence for the ground-truth class. While each metric captures a unique aspect of sample difficulty, they are not mutually exclusive. For instance, T3 directly quantifies the prediction error, which is often correlated with the other terms. However, their combination provides a more holistic assessment. Together, these metrics prioritize moderately difficult samples situated near class decision boundaries, which correspond to moderately forgettable cases that play a critical role in enhancing representation robustness. These metrics are computed as:

$$\mathcal{T}_1^b = \frac{1}{CHW} \sum_{c,h,w} \|\mathbf{f}_{b,c,h,w} - \boldsymbol{\mu}_c\|_2^2 \tag{7}$$

$$\mathcal{T}_2^b = \frac{1}{C(C-1)HW} \sum_{c \neq c', h, w} \frac{1}{\|\mathbf{f}_{b,c,h,w} - \boldsymbol{\mu}_{c'}\|_2^2 + \epsilon} \tag{8}$$

$$\mathcal{T}_3^b = \frac{1}{CHW} \sum_{c,h,w} \|\mathbf{p}_{b,c,h,w} - \mathbf{m}_{b,c,h,w}\|_2^2 \tag{9}$$

$$\mathcal{T}_4^b = 1 - \frac{1}{CHW} \sum_{c,h,w} \mathbf{p}_{b,c,h,w} \tag{10}$$

where $\epsilon$ is a small positive constant to ensure numerical stability. The four terms are normalized by the number of pixels to bring them to a comparable scale. The unified difficulty score is an unweighted sum, reflecting a balanced consideration of these complementary aspects:

$$\text{Score}^b = \mathcal{T}_1^b + \mathcal{T}_2^b + \mathcal{T}_3^b + \mathcal{T}_4^b \tag{11}$$

This scoring strategy reliably identifies moderately forgettable samples that are clinically informative. By integrating feature geometry (T1, T2) with prediction-based analysis (T3, T4), it provides a more comprehensive difficulty measure than confidence scores alone, enabling efficient and controllable hard positive mining.

### 3.3 FORGETTABLE SAMPLE BANK

Building on the difficulty scores derived from Prototype-Based Scoring, we maintain an Online Forgettable Sample Bank to mitigate repeated forgetting and enable curriculum-inspired continual learning. Curriculum learning Bengio et al. (2009); Fan et al. (2017) progressively focuses on samples of varying difficulty to improve model robustness and generalization. By targeting moderately forgettable samples, which reside near decision boundaries and capture key variations, the bank ensures that challenging cases are revisited systematically while maintaining training stability.

Focusing on these samples enables the model to better distinguish clinically relevant hard positives, leading to improved robustness and reduced false negative rates. For each batch of size $B$, we identify the top $n_{\text{hard}} = \lfloor B \cdot \rho \rfloor$ hardest samples using:

$$\mathcal{I}_{\text{hard}} = \text{top-k}(\text{Score}^b, n_{\text{hard}}) \tag{12}$$

These samples are stored in a memory bank $\text{M}_{\text{bank}}$ of size $P \times B$, which is updated by replacing $n_{\text{hard}}$ randomly chosen entries with new hard samples:

$$\text{M}_{\text{bank}} \leftarrow \text{RS}(\text{M}_{\text{bank}}, n_{\text{hard}}) \cup \mathcal{I}_{\text{hard}} \tag{13}$$

Here, $\text{RS}(\text{M}_{\text{bank}}, n_{\text{hard}})$ denotes randomly sampling $n_{\text{hard}}$ entries from the memory bank for replacement. Random replacement avoids temporal bias and maintains diversity, while the multi-dimensional screening via Prototype-Based Scoring ensures that stored samples are both informative and representative. This design prevents the accumulation of redundant or uninformative data and strategically focuses on hard positives that contribute most to robust representation learning.

To further enhance learning, every $P$ main batch, a mini-batch is sampled from the memory bank for replay:

$$\text{batch}_{\text{mini}} = \text{RS}(\text{M}_{\text{bank}}, B) \tag{14}$$

The replay mechanism samples mini-batches from the memory bank for repeated training, increasing exposure to ambiguous and abnormal regions. By integrating semantic cues from Text-Guided Fusion, the model can better utilize replayed samples to distinguish subtle lesion features from background noise, especially in visually challenging areas.

In summary, the Online Forgettable Sample Bank leverages semantic guidance and difficulty-aware replay to enhance the model's discrimination of difficult regions and reduce false negatives in challenging segmentation tasks.

## 4 EXPERIMENTS AND RESULTS

### 4.1 DATASETS AND IMPLEMENTATION DETAILS

We evaluate our method on five public medical imaging datasets spanning diverse anatomical structures and imaging modalities. The **Kidney and Kidney Tumor Segmentation (KiTS) 2023** dataset

Heller et al. (2023) includes 489 CT scans. The **Brain Tumor Segmentation (BraTS) 2020** dataset Menze et al. (2014) contains 369 multimodal MRI cases. The **Automated Cardiac Diagnosis Challenge (ACDC)** dataset Bernard et al. (2018) provides 100 cardiac MRI cases. The **Fast and Low GPU Memory Abdominal Organ Segmentation (FLARE) 2021** dataset Ma et al. (2022) consists of 361 abdominal CT scans. The **Prostate MRI Image Segmentation (PROMISE) 2012** dataset Litjens et al. (2014) contains 50 MRI cases.

All datasets are split into training, validation, and test sets in a 4:1:1 ratio. To standardize the training protocol, models are trained on 2D patches with a batch size of 32 and a patch size of $256 \times 256$, using stochastic gradient descent (initial learning rate: 0.01) on an NVIDIA RTX 4090 GPU. The loss function is cross-entropy loss, and performance is evaluated using the Dice similarity coefficient (DSC), 95% Hausdorff distance (HD95), and sensitivity (Sen). Our approach adopts a 2D UNet Ronneberger et al. (2015) as the backbone. To ensure a fair comparison of the proposed sample mining strategy, all baseline models, including those originally designed for 3D data (e.g., UNETR, nnU-Net), were adapted to the same 2D patch-based framework. This ensures performance differences reflect the core mining mechanism rather than architectural or implementation variations.

## 4.2 RESULTS AND ANALYSIS

Table 1: Comparison of DSC↑ and Sensitivity (Sens↑) results across different datasets. **Bold** indicates the best results, *italic* indicates the second-best results.

| Dataset | Target | UNETR | | MambaUNet | | AttentionUNet | | nnUNet | | nnUNet+TL | | nnUNet+BL | | nnUNet+BDL | | nnUNet+FL | | Ours | |
|---|---|---|---|---|---|---|---|---|---|---|---|---|---|---|---|---|---|---|---|
| | | DSC↑ | Sens↑ | DSC↑ | Sens↑ | DSC↑ | Sens↑ | DSC↑ | Sens↑ | DSC↑ | Sens↑ | DSC↑ | Sens↑ | DSC↑ | Sens↑ | DSC↑ | Sens↑ | DSC↑ | Sens↑ |
| KiTS2023 | Kidney and Masses | 0.908 | 0.889 | 0.908 | 0.897 | 0.888 | 0.878 | 0.912 | 0.899 | 0.908 | 0.888 | 0.912 | 0.898 | *0.913* | *0.900* | 0.909 | 0.897 | **0.917** | **0.906** |
| | Kidney Mass | 0.663 | 0.708 | 0.699 | 0.707 | 0.625 | 0.611 | 0.703 | 0.702 | 0.700 | 0.674 | 0.707 | 0.699 | *0.709* | 0.700 | 0.705 | **0.709** | **0.715** | 0.707 |
| | Tumor | 0.626 | 0.690 | 0.681 | 0.692 | 0.618 | 0.658 | 0.687 | 0.695 | 0.680 | 0.666 | 0.683 | 0.689 | *0.689* | *0.697* | 0.683 | **0.699** | **0.693** | 0.696 |
| | Avg. | 0.733 | 0.763 | 0.762 | 0.765 | 0.710 | 0.716 | 0.768 | 0.765 | 0.763 | 0.742 | 0.767 | 0.762 | *0.771* | *0.766* | 0.765 | 0.768 | **0.775** | **0.769** |
| BraTS2020 | Whole Tumor | 0.912 | 0.903 | 0.916 | 0.897 | 0.910 | **0.913** | **0.919** | 0.901 | 0.875 | 0.875 | 0.910 | 0.905 | 0.910 | 0.905 | 0.915 | 0.892 | *0.918* | *0.905* |
| | Tumor Core | **0.846** | **0.814** | 0.823 | 0.812 | 0.843 | 0.811 | 0.836 | 0.812 | 0.832 | 0.809 | 0.837 | 0.831 | 0.832 | 0.834 | 0.833 | 0.813 | *0.839* | **0.840** |
| | Enhancing Tumor | 0.796 | 0.819 | 0.781 | 0.795 | 0.789 | *0.815* | 0.794 | 0.803 | 0.785 | 0.784 | 0.797 | 0.812 | 0.791 | 0.810 | 0.781 | 0.798 | **0.800** | 0.805 |
| | Avg. | *0.851* | *0.846* | 0.840 | 0.835 | 0.847 | 0.847 | 0.850 | 0.839 | 0.831 | 0.823 | 0.848 | *0.850* | 0.844 | 0.850 | 0.843 | 0.835 | **0.852** | **0.850** |
| ACDC | Right Ventricle | 0.909 | 0.891 | 0.901 | 0.895 | 0.898 | 0.891 | *0.911* | 0.889 | 0.908 | 0.888 | 0.908 | 0.890 | **0.915** | 0.896 | 0.901 | 0.892 | 0.909 | **0.903** |
| | Myocardium | 0.907 | 0.920 | *0.911* | *0.927* | *0.911* | *0.927* | 0.906 | 0.921 | 0.907 | 0.915 | 0.908 | 0.919 | 0.907 | 0.918 | 0.906 | 0.911 | **0.912** | 0.924 |
| | Left Ventricle | 0.946 | 0.961 | 0.947 | 0.955 | 0.945 | 0.957 | 0.947 | 0.959 | 0.947 | 0.962 | 0.949 | *0.962* | 0.947 | **0.962** | 0.945 | 0.959 | **0.952** | **0.962** |
| | Avg. | 0.921 | 0.924 | 0.920 | 0.926 | 0.918 | 0.925 | 0.921 | 0.923 | 0.921 | 0.922 | 0.922 | 0.924 | *0.923* | 0.925 | 0.918 | 0.921 | **0.925** | **0.930** |
| PROMISE2012 | Prostate | 0.851 | 0.831 | 0.860 | 0.833 | 0.860 | 0.845 | 0.872 | *0.853* | 0.852 | 0.815 | 0.853 | 0.827 | 0.862 | 0.823 | 0.863 | 0.842 | **0.883** | **0.869** |
| FLARE2021 | Liver | 0.969 | 0.968 | 0.975 | 0.972 | 0.968 | 0.968 | 0.976 | 0.971 | 0.975 | 0.970 | 0.977 | 0.972 | *0.978* | 0.972 | 0.975 | *0.973* | **0.982** | **0.982** |
| | Kidney | 0.958 | 0.967 | 0.962 | *0.971* | 0.956 | 0.965 | 0.964 | 0.969 | 0.963 | 0.968 | *0.965* | 0.969 | **0.966** | 0.970 | 0.962 | 0.970 | 0.965 | **0.974** |
| | Spleen | 0.948 | 0.949 | 0.952 | *0.953* | 0.946 | 0.947 | 0.954 | 0.950 | 0.953 | 0.950 | 0.955 | 0.951 | *0.963* | 0.951 | 0.953 | 0.952 | **0.975** | **0.976** |
| | Pancreas | 0.768 | 0.783 | 0.775 | 0.791 | 0.766 | 0.780 | 0.782 | 0.795 | 0.780 | 0.789 | 0.785 | 0.794 | *0.796* | 0.798 | 0.776 | *0.802* | **0.808** | **0.807** |
| | Avg. | 0.911 | 0.917 | 0.916 | 0.922 | 0.909 | 0.915 | 0.919 | 0.921 | 0.918 | 0.919 | 0.920 | 0.922 | *0.926* | 0.923 | 0.916 | *0.924* | **0.932** | **0.935** |

Table 2: Comparison of HD95↓ results across different datasets. **Bold** indicates the best results, *italic* indicates the second-best results.

| Dataset | Target | UNETR | MambaUNet | AttentionUNet | nnUNet | nnUNet+TL | nnUNet+BL | nnUNet+BDL | nnUNet+FL | Ours |
|---|---|---|---|---|---|---|---|---|---|---|
| KiTS2023 | Kidney and Masses | 18.6792 | 15.5281 | 16.0463 | 15.8870 | 16.3882 | 15.4733 | **14.8362** | 15.3346 | *14.9658* |
| | Kidney Mass | 63.5917 | *43.5350* | 53.5642 | 45.1208 | 43.7024 | 44.8564 | 46.1955 | 44.9539 | **43.2246** |
| | Tumor | 67.9381 | 54.6875 | 73.1334 | 55.4069 | 56.0211 | 55.7482 | **53.6599** | *53.8435* | 55.1206 |
| | Avg. | 50.0697 | *37.9168* | 47.5813 | 38.8049 | 38.7039 | 38.6926 | 38.2305 | 38.0440 | **37.7703** |
| BraTS2020 | Whole Tumor | 1.9825 | 1.8586 | 2.3947 | 1.9208 | 1.9301 | **1.7999** | *1.8025* | 1.8238 | 1.9473 |
| | Tumor Core | 3.9111 | 4.4228 | 4.5787 | 4.0340 | 3.5134 | 3.4437 | 3.5286 | *3.2747* | **3.2612** |
| | Enhancing Tumor | 3.8810 | 3.5274 | 3.8636 | **3.2727** | *3.3639* | 3.4003 | 3.3654 | 4.3936 | 3.3840 |
| | Avg. | 3.2582 | 3.2696 | 3.6123 | 3.0758 | 2.9358 | *2.8813* | 2.8988 | 3.1640 | **2.8642** |
| ACDC | Right Ventricle | *0.6783* | 0.7398 | 0.9652 | 0.7086 | 0.7357 | 0.7111 | 0.6751 | 0.7746 | **0.5676** |
| | Myocardium | 0.9235 | 0.8921 | 0.9085 | 0.8853 | 0.8765 | 0.8648 | *0.8527* | 0.8974 | **0.8432** |
| | Left Ventricle | 0.4671 | *0.3636* | 0.4545 | **0.3606** | 0.4890 | 0.3939 | 0.4368 | 0.4368 | 0.4242 |
| | Avg. | 0.6896 | 0.6652 | 0.7761 | *0.6515* | 0.7004 | 0.6566 | 0.6549 | 0.7029 | **0.6117** |
| PROMISE2012 | Prostate | 1.9650 | 1.8018 | 1.9231 | 1.6342 | 1.7165 | 1.9079 | *1.6339* | 1.7500 | **1.5000** |
| FLARE2021 | Liver | 2.2436 | 2.0065 | 2.1657 | 1.9877 | 1.9532 | 1.9325 | *1.8746* | 2.1118 | **1.6667** |
| | Kidney | 0.9877 | 0.8936 | 0.9425 | 0.8824 | 0.9125 | 0.8943 | *0.8722* | 0.9581 | **0.5793** |
| | Spleen | 11.5674 | 10.9649 | 11.3246 | 10.8526 | 10.7365 | 10.6258 | 10.5388 | *10.4413* | **3.1437** |
| | Pancreas | 8.8965 | 8.4376 | 8.6543 | 8.2466 | 8.1254 | 8.0246 | 7.8873 | *7.4739* | **3.7504** |
| | Avg. | 5.9238 | 5.5756 | 5.7718 | 5.4923 | 5.4319 | 5.3693 | 5.2932 | *5.2463* | **2.2850** |

**Quantitative Results.** To validate our method, we conduct comparisons against two categories of baselines. First, we benchmark against leading architectures including Transformer-based UNETR Hatamizadeh et al. (2022), Mamba-based MambaUNet Wang et al. (2024b), and Attention U-Net Oktay et al. (2018) to demonstrate competitive performance. Second, to directly evaluate our hard-sample mining contribution, we compare against established difficulty-measuring strategies using nnU-Net Isensee et al. (2021) as a strong baseline. The baselines include hard-sample-oriented losses: Tversky Loss (TL) Salehi et al. (2017), Boundary Loss (BL) Kervadec et al. (2019), BoundaryDoULoss (BDL) Sun et al. (2023), and Focal Loss (FL) Lin et al. (2017), representing alternative

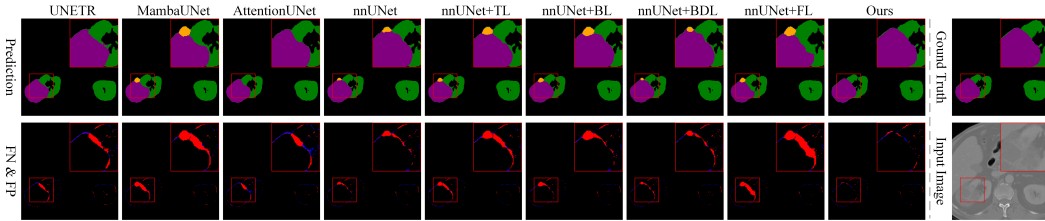

Figure 3: Segmentation results comparison on KiTS2023 dataset. First row: predictions for Kidney (green), Tumor (purple), and Cyst (orange). Second row: error maps with false negatives (FN, red) and false positives (FP, blue). Input CT images and ground truth are shown on the right.

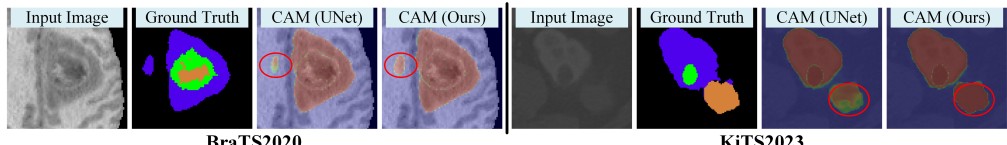

Figure 4: Class activation maps (CAM) comparison on KiTS2023 and BraTS2020 datasets, highlighting regions most influential to model decisions (blue: low activation, red: high activation).

paradigms for measuring sample difficulty. All hard-sample mining methods are implemented on a unified nnU-Net backbone to ensure fair comparison. The core question is not architectural but strategic: how to define sample difficulty. Baseline methods leverage implicit difficulty heuristics from low-level cues (prediction confidence, geometric location), while our method introduces explicit semantic priors to define difficulty based on feature-space coherence. CLIP serves exclusively as a tool to instantiate this semantic criterion without providing additional features to the segmentation backbone. This setup enables direct comparison of a fundamental question: is sample difficulty better defined by implicit, output-based heuristics or explicit, external semantic priors?

As shown in Table 1, our method consistently achieves superior DSC and sensitivity across diverse datasets and anatomical targets. Table 2 further demonstrates that our model yields favorable boundary accuracy (HD95), indicating strong performance in both overlap and positive region detection. Notably, our DSC is slightly lower than that of nnU-Net and certain variants on the Whole Tumor region. This occurs because Whole Tumor segmentation includes large, well-defined tumor areas where overlap-based optimization (as in nnU-Net) is highly effective. Methods like nnU-Net+BL and nnU-Net+BDL, which explicitly optimize for boundary localization, naturally excel in such scenarios with clear volumetric boundaries. In contrast, our approach leverages semantic priors to guide hard positive identification over simple volumetric overlap, making it particularly effective for challenging targets like Tumor Core and Enhancing Tumor where boundary ambiguity is more pronounced and purely visual cues are insufficient. This is further evidenced by our leading performance on Tumor Core, Enhancing Tumor, and other challenging targets. Across most scenarios, the balance between sensitivity and HD95 achieved by our method underscores its strong generalizability.

**Qualitative Results.** Fig. 3 shows our method delivers precise delineation of complex boundaries on KiTS2023. Comparison methods incorrectly classify kidney and tumor regions as cysts where no cysts exist in the ground truth, indicating class confusion between visually similar structures. Error maps show reduced false negatives and false positives, with greatest gains at organ-tumor boundaries. This occurs because comparison methods rely on visual features alone, causing misclassification of low-contrast regions. Our CLIP semantic guidance provides discriminative information beyond visual appearance, while hard positive mining targets ambiguous boundary regions, preventing kidney-to-cyst and tumor-to-cyst misclassifications.

CAM visualizations in Fig. 4 show that the baseline UNet exhibits weak, scattered activations in hard positive regions, whereas our method generates concentrated activations within target boundaries. Our Forgettable Sample Bank repeatedly trains on specific hard regions, and Text-Guided

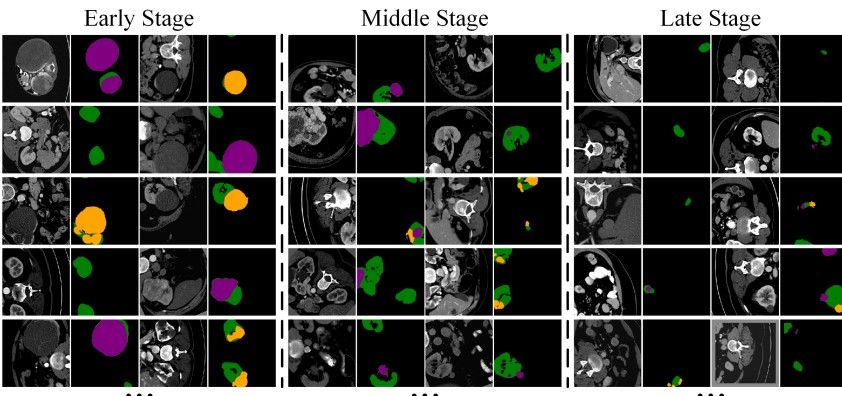

Figure 5: Dynamic evolution of the Online Forgettable Sample Bank on KiTS2023. As training progresses, the bank shifts from storing large, well-defined anatomical structures to focusing on moderately forgettable samples like irregularly shaped, low-contrast, or small fragmented regions.

Table 3: Ablation study results on BraTS2020 and FLARE2021 showing contribution of each module. Text-Guided Fusion (TGF), Prototype-Based Scoring (PBS), Forgettable Sample Bank (FSB).

| Module | | | BraTS2020 | | | FLARE2021 | | |
|---|---|---|---|---|---|---|---|---|
| TGF | PBS | FSB | DSC↑ | HD95↓ | Sens↑ | DSC↑ | HD95↓ | Sens↑ |
| × | × | × | 0.8471 | 3.3438 | 0.8391 | 0.9196 | 5.6755 | 0.9201 |
| × | ✓ | ✓ | 0.8511 | 2.8974 | 0.8463 | 0.9228 | 4.1802 | 0.9244 |
| ✓ | × | × | 0.8472 | 2.9538 | 0.8420 | 0.9218 | 4.8727 | 0.9221 |
| ✓ | ✓ | ✓ | 0.8520 | 2.8642 | 0.8502 | 0.9321 | 2.2850 | 0.9352 |

Fusion provides semantic constraints focusing on class-relevant features, enabling confident activation on true positives while suppressing background noise. This is further supported by the dynamic evolution of our sample bank, which adaptively focuses on such challenging regions as training progresses, as shown in Fig. 5.

**Ablation Analysis.** Ablation results (Table 3) on multiple datasets (e.g., FLARE2021) demonstrate the consistent contribution of each module. The baseline UNet provides the reference. Adding Text-Guided Fusion improves class discrimination and boundary accuracy, as CLIP semantic embeddings offer complementary semantic cues. Prototype-Based Scoring and Forgettable Sample Bank are evaluated jointly, as difficulty-based sample selection must operate on a dynamically updated sample pool to be effective. Adding Prototype-Based Scoring enhances sample selection, enabling the model to focus on optimal training difficulty. The Forgettable Sample Bank increases sensitivity by providing a more consistent supply of informative hard positives. The full combination yields the best overall performance. More detailed ablation studies and hyperparameter settings are provided in the Appendix.

## 5  CONCLUSION

We present an end-to-end framework that addresses the critical challenge of distinguishing visually similar anatomical structures in medical image segmentation by mining moderately forgettable samples through CLIP semantic guidance, prototype scoring, and a forgettable sample bank to prevent misclassification and reduce false negatives at organ boundaries. Experiments demonstrate consistent improvements, with ablation studies confirming each module's effectiveness in mining moderately forgettable samples for addressing visual confusion.

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

## A APPENDIX

This appendix provides supplementary materials, including detailed ablation analyses of key model components such as prompt templates, sample selection mechanisms, memory bank management strategies, individual scoring metrics, and hyperparameters. A statement on the use of Large Language Models (LLMs) in preparing this manuscript is included in Section A.1.

### A.1 STATEMENT ON LLM USAGE

In accordance with conference guidelines, we report the use of a Large Language Model (LLM) during the preparation of this manuscript. The LLM's role was strictly limited to improving the clarity, conciseness, and grammatical correctness of the text. All scientific contributions—including research ideation, experimental design, and data analysis—were performed by the human authors, who have reviewed, edited, and take full responsibility for the content of this paper.

### A.2 DATA PREPROCESSING

We applied a standard preprocessing pipeline to all datasets to ensure consistency and stable model training. This pipeline was used for all methods to guarantee a fair comparison.

**Image Resampling and Spacing Normalization:** All images were resampled to a unified isotropic voxel spacing. This spacing was set to the median voxel spacing of the dataset. Trilinear interpolation was used for intensity images, while nearest-neighbor interpolation was applied to corresponding segmentation masks to preserve discrete label integrity.

**Intensity Normalization:** For CT images, Hounsfield Unit (HU) values were first clipped to a clinically relevant range of -1000 to 400 HU. Subsequently, Z-score normalization (subtracting the mean and dividing by the standard deviation) was applied. For MRI data, we applied Z-score standardization exclusively to foreground voxels (i.e., non-zero pixels) to prevent background regions from skewing normalization statistics.

**Data Augmentation Strategies:** During training, a suite of data augmentation techniques was employed to enhance model robustness and generalization. These techniques included gamma transformation, additive Gaussian noise, Gaussian blurring, and brightness adjustment.

### A.3 THEORETICAL MOTIVATION

Our proposed hard-sample mining framework is theoretically grounded in metric learning and probabilistic principles. The objective is to identify samples that are ambiguous or poorly represented within the learned embedding space.

**Prototypes as Estimators of Class-Conditional Distributions.** We model the semantically-enhanced visual features, $\mathbf{f}_{\text{output}}$, as samples drawn from a mixture of class-conditional distributions. Specifically, for a pixel belonging to class $c$, its feature vector is a sample from a distribution $p(\mathbf{f}|y = c)$. The class prototypes, $\boldsymbol{\mu}_c$, which are computed as the running average of features for class $c$, serve as online estimators of the distribution means:

$$\boldsymbol{\mu}_c \approx \mathbb{E}[\mathbf{f}|y = c] \tag{15}$$

The Exponential Moving Average (EMA) update, $\boldsymbol{\mu}_c \leftarrow \beta\boldsymbol{\mu}_c + (1 - \beta)\boldsymbol{\mu}_c^{\text{current}}$, acts as a low-pass filter, providing a stable, noise-reduced estimate of the true class centroids over the non-stationary training trajectory.

**Difficulty Score as a Proxy for Semantic Uncertainty.** A sample is considered "hard" if its feature representations are inconsistent with this learned probabilistic structure. Our multi-metric score, $\mathrm{Score}^b$, is designed to approximate this feature inconsistency, termed *semantic uncertainty*. A high score identifies samples where: (a) The intra-class feature variance is high, indicating that the sample's features are far from their own class prototype $\boldsymbol{\mu}_c$. This corresponds to low likelihood under the estimated class-conditional distribution $p(\mathbf{f}|y = c)$. (b) The inter-class feature distance is low, meaning the sample's features are close to one or more incorrect class prototypes $\boldsymbol{\mu}_{c' \neq c}$. This signifies high ambiguity and potential for misclassification in the embedding space. Therefore, the score serves as a principled measure of a sample's deviation from an ideally separated class manifold, rather than an ad-hoc heuristic.

**Memory Bank as Online Importance Sampling.** Standard mini-batch SGD assumes that samples are drawn i.i.d. from the training distribution, an assumption that often fails for rare and correlated hard samples. The Forgettable Sample Bank, $\mathrm{M_{bank}}$, combined with periodic replay, can be viewed as a form of online importance sampling that addresses this issue. The bank constructs an empirical approximation of the true distribution of hard samples, $p_{\mathrm{hard}}(\mathbf{x}, \mathbf{y})$. By replaying samples from this bank, we correct the uniform sampling assumption of SGD and dedicate additional gradient updates to the most informative, high-uncertainty regions of the data distribution. This process is intended to accelerate convergence and enhance generalization.

## A.4 ALGORITHM PSEUDOCODE

Algorithm 1 details the training procedure of our hard positive mining framework, which consists of three main steps for each training batch:

1. **Scoring and Bank Management:** Following a forward pass that fuses visual features with CLIP-based semantic guidance, we compute and update class prototypes using an EMA for temporal stability. A multi-metric difficulty score is then calculated for each sample based on these prototypes. The hardest samples identified in the batch are used to update an online memory bank via random replacement.

2. **Standard Optimization:** A standard segmentation loss (e.g., Cross-Entropy and Dice) is computed on the current batch. This loss constitutes the primary component of the total optimization objective.

3. **Replay-Based Reinforcement:** Periodically (every $F$ iterations), a mini-batch of hard samples is drawn from the memory bank. A separate forward pass computes a replay loss for this batch, which is added to the main loss. The model parameters are then updated based on the gradients from this combined objective, ensuring that the model reinforces its learning on the most informative and challenging examples identified over time.

## A.5 ANALYSIS OF PROMPT TEMPLATES

Table 4: Impact of different CLIP prompt templates on BraTS2020 and FLARE2021 segmentation performance.

| Prompt Template | BraTS2020 | | | FLARE2021 | | |
|---|---|---|---|---|---|---|
| | DSC↑ | HD95↓ | Sens↑ | DSC↑ | HD95↓ | Sens↑ |
| A photo of a [object]. | 0.8502 | 3.0536 | 0.8452 | 0.9290 | 3.2505 | 0.9280 |
| There is [object] in this magnetic resonance imaging. | 0.8477 | 2.9513 | 0.8428 | 0.9270 | 3.0101 | 0.9260 |
| A magnetic resonance imaging of a [object]. | 0.8520 | 2.8642 | 0.8502 | 0.9324 | 2.2850 | 0.9353 |

To effectively leverage CLIP's semantic knowledge, text prompts must be aligned with the medical imaging domain. We tested three templates, with results presented in Table 4. The generic prompt "A photo of a [object]" yielded strong but suboptimal results. A more descriptive yet complex prompt, "There is [object] in this magnetic resonance imaging," slightly degraded performance. In contrast, the best-performing template, "A magnetic resonance imaging of a [object]," achieved the highest DSC (0.8520), highest sensitivity (0.8502), and lowest HD95 (2.8642) on BraTS2020, with similarly superior results on FLARE2021 (DSC: 0.9324, HD95: 2.2850, Sens: 0.9353). This finding

---

**Algorithm 1** Training Procedure with Prototype-Based Hard Positive Mining

---

**Require:** Segmentation model $f_\theta$, frozen CLIP text encoder $\Phi_{\text{CLIP}}$, training data loader $\mathcal{D}$.
**Require:** Hyperparameters: learning rate $\eta$, EMA coefficient $\beta$, mining ratio $\rho$, bank capacity $P$, replay frequency $F$.
**Ensure:** Trained model parameters $\theta$.
 1: Initialize model parameters $\theta$, class prototypes $\boldsymbol{\mu}_c \leftarrow \mathbf{0}$, Forgettable Sample Bank $M_{\text{bank}} \leftarrow \emptyset$.
 2: Pre-compute text embeddings $\mathbf{f}_{\text{CLIP}}$ for all classes using $\Phi_{\text{CLIP}}$.
 3: `iter_count` $\leftarrow 0$.
 4: **for** each batch $\{\mathbf{x}, \mathbf{y}\}$ in $\mathcal{D}$ **do**
 5: $\quad$ `iter_count` $\leftarrow$ `iter_count` $+ 1$
 6: $\quad$ **// Forward Pass with Text-Guided Fusion**
 7: $\quad$ $\mathbf{f}_{\text{visual}} \leftarrow \text{Encoder}_\theta(\mathbf{x})$
 8: $\quad$ $\mathbf{f}' \leftarrow \text{MultiHeadAttention}(\mathbf{f}_{\text{visual}}, \mathbf{f}_{\text{CLIP}})$
 9: $\quad$ $\mathbf{f}_{\text{output}} \leftarrow \text{Decoder}_\theta(\mathbf{f}')$ $\qquad\qquad\qquad\qquad$ ▷ Obtain semantically-enhanced features
10: $\quad$ **// Prototype-Based Difficulty Scoring**
11: $\quad$ $\boldsymbol{\mu}_c^{\text{current}} \leftarrow \frac{\sum \mathbf{f}_{\text{output}} \cdot \mathbb{I}[\mathbf{y}=c]}{\sum \mathbb{I}[\mathbf{y}=c]}$ $\qquad\qquad\qquad$ ▷ Compute current batch prototypes
12: $\quad$ $\boldsymbol{\mu}_c \leftarrow \beta \boldsymbol{\mu}_c + (1-\beta) \boldsymbol{\mu}_c^{\text{current}}$ $\qquad\qquad\quad$ ▷ Update global prototypes via EMA
13: $\quad$ **for** $b = 1, \dots, B$ **do** $\qquad\qquad\qquad\qquad$ ▷ Iterate over each sample in the batch
14: $\quad\quad$ Compute metrics $\mathcal{T}_1^b, \mathcal{T}_2^b, \mathcal{T}_3^b, \mathcal{T}_4^b$ using $\mathbf{f}_{\text{output}}^b, \mathbf{y}^b, \boldsymbol{\mu}_c$.
15: $\quad\quad$ $\text{Score}^b \leftarrow \mathcal{T}_1^b + \mathcal{T}_2^b + \mathcal{T}_3^b + \mathcal{T}_4^b$.
16: $\quad$ **end for**
17: $\quad$ **// Online Forgettable Sample Bank Update**
18: $\quad$ $n_{\text{hard}} \leftarrow \lfloor B \cdot \rho \rfloor$.
19: $\quad$ $\mathcal{I}_{\text{hard}} \leftarrow \text{top-k}(\{\text{Score}^b\}_{b=1}^B, n_{\text{hard}})$ $\qquad\qquad$ ▷ Identify indices of hardest samples
20: $\quad$ $(\mathbf{x}_{\text{hard}}, \mathbf{y}_{\text{hard}}) \leftarrow (\mathbf{x}[\mathcal{I}_{\text{hard}}], \mathbf{y}[\mathcal{I}_{\text{hard}}])$.
21: $\quad$ Update $M_{\text{bank}}$ by replacing $n_{\text{hard}}$ random entries with $(\mathbf{x}_{\text{hard}}, \mathbf{y}_{\text{hard}})$.
22: $\quad$ **// Model Optimization and Replay**
23: $\quad$ $\mathcal{L}_{\text{main}} \leftarrow \text{SegmentationLoss}(\mathbf{f}_{\text{output}}, \mathbf{y})$.
24: $\quad$ $\mathcal{L}_{\text{total}} \leftarrow \mathcal{L}_{\text{main}}$.
25: $\quad$ **if** `iter_count` mod $F = 0$ AND $|M_{\text{bank}}| \geq B$ **then**
26: $\quad\quad$ $(\mathbf{x}_{\text{replay}}, \mathbf{y}_{\text{replay}}) \leftarrow \text{RandomSample}(M_{\text{bank}}, B)$.
27: $\quad\quad$ $\mathbf{f}_{\text{output}}^{\text{replay}} \leftarrow f_\theta(\mathbf{x}_{\text{replay}})$ $\qquad\qquad\qquad$ ▷ Forward pass on replayed samples
28: $\quad\quad$ $\mathcal{L}_{\text{replay}} \leftarrow \text{SegmentationLoss}(\mathbf{f}_{\text{output}}^{\text{replay}}, \mathbf{y}_{\text{replay}})$.
29: $\quad\quad$ $\mathcal{L}_{\text{total}} \leftarrow \mathcal{L}_{\text{total}} + \mathcal{L}_{\text{replay}}$.
30: $\quad$ **end if**
31: $\quad$ Update $\theta$ based on gradients from $\mathcal{L}_{\text{total}}$.
32: **end for**

---

demonstrates that a prompt that is both domain-specific and structurally concise provides the most effective semantic embeddings for guiding our difficulty scoring.

A.6 ANALYSIS OF MEMORY BANK MANAGEMENT

Table 5: Comparison of Forgettable Sample Bank management strategies on BraTS2020 and FLARE2021.

| Strategy | BraTS2020 | | | FLARE2021 | | |
|---|---|---|---|---|---|---|
| | DSC↑ | HD95↓ | Sens↑ | DSC↑ | HD95↓ | Sens↑ |
| FIFO | 0.8510 | 2.9128 | 0.8487 | 0.9290 | 3.1502 | 0.9280 |
| Score-based Sampling | 0.8515 | 2.9057 | 0.8579 | 0.9300 | 2.4505 | 0.9310 |
| Random Sampling | 0.8520 | 2.8642 | 0.8502 | 0.9324 | 2.2850 | 0.9353 |

The bank update strategy is critical for maintaining sample diversity. As shown in Table 5, we compare three strategies. First-In-First-Out (FIFO) serves as a simple baseline, while score-based replacement offers marginal improvements. However, our proposed random sampling strategy consistently yields the best results. We hypothesize that deterministic methods like FIFO and score-

based replacement can introduce sampling bias, causing the bank to be dominated by common types of hard samples. In contrast, random sampling avoids this issue by maintaining a more diverse collection of historical hard samples, thereby promoting more robust and generalizable model training.

## A.7 ANALYSIS OF SAMPLE SELECTION STRATEGY

Table 6: Comparison of sample selection strategies for Prototype-Based Scoring on BraTS2020 and FLARE2021.

| Selection Strategy | BraTS2020 | | | FLARE2021 | | |
|---|---|---|---|---|---|---|
| | DSC↑ | HD95↓ | Sens↑ | DSC↑ | HD95↓ | Sens↑ |
| Loss-based | 0.8493 | 2.9529 | 0.8485 | 0.9280 | 3.3001 | 0.9270 |
| Random | 0.8481 | 3.0188 | 0.8443 | 0.9260 | 3.5506 | 0.9250 |
| Our Score | 0.8520 | 2.8642 | 0.8502 | 0.9324 | 2.2850 | 0.9353 |

To validate our multi-dimensional scoring mechanism, we compared it against two common baselines: random selection and selection based on segmentation loss. As shown in Table 6, random selection is the least effective, as it fails to target difficult samples. Loss-based selection offers improvement but is consistently surpassed by our multi-dimensional scoring. Our method achieves the highest performance on both datasets, confirming the superiority of a holistic difficulty measure. While loss primarily reflects prediction error, our feature-space metrics (intra-class consistency and inter-class distinction) identify challenges related to boundary ambiguity and semantic confusion. This combination provides a more holistic assessment of sample difficulty.

## A.8 ANALYSIS OF INDIVIDUAL SCORING COMPONENTS

Table 7: Ablation study of individual scoring metrics in Prototype-Based Scoring on BraTS2020 and FLARE2021.

| $\mathcal{T}_1$ | $\mathcal{T}_2$ | $\mathcal{T}_3$ | $\mathcal{T}_4$ | BraTS2020 | | | FLARE2021 | | |
|---|---|---|---|---|---|---|---|---|---|
| | | | | DSC↑ | HD95↓ | Sens↑ | DSC↑ | HD95↓ | Sens↑ |
| ✓ | × | × | × | 0.8481 | 2.9855 | 0.8458 | 0.9270 | 3.6509 | 0.9260 |
| × | ✓ | × | × | 0.8468 | 3.0522 | 0.8416 | 0.9260 | 3.9008 | 0.9250 |
| × | × | ✓ | × | 0.8514 | 2.8972 | 0.8489 | 0.9310 | 2.3503 | 0.9300 |
| × | × | × | ✓ | 0.8456 | 3.0746 | 0.8419 | 0.9250 | 4.0506 | 0.9250 |
| ✓ | ✓ | ✓ | ✓ | 0.8520 | 2.8642 | 0.8502 | 0.9324 | 2.2850 | 0.9353 |

We conducted an ablation study to evaluate the contribution of each of the four scoring components ($\mathcal{T}_1$–$\mathcal{T}_4$). Table 7 presents the performance of each metric individually versus their combination. While $\mathcal{T}_3$ (prediction deviation) shows strong individual performance, the synergistic combination of all four metrics yields superior results. This outcome validates our hypothesis that a comprehensive difficulty measure requires integrating multiple perspectives. Specifically, $\mathcal{T}_3$ and $\mathcal{T}_4$ assess difficulty in the **prediction space** (error and uncertainty), whereas $\mathcal{T}_1$ and $\mathcal{T}_2$ evaluate difficulty in the **feature space** (intra-class dispersion and inter-class ambiguity). The feature-space metrics are crucial for identifying samples that are semantically confusing, even if their prediction error is not maximal. By integrating these distinct views, our framework achieves a more robust identification of hard samples.

## A.9 HYPERPARAMETER SENSITIVITY ANALYSIS

We analyzed the model's sensitivity to two key hyperparameters: the mining ratio ($\rho$) and the memory bank capacity ($P$). As shown in Table 8, performance is consistently optimal with $\rho = 0.1$ and $P = 10$. For the mining ratio $\rho$, a value that is too low provides insufficient exposure to hard samples, while a value that is too high can destabilize training. For the bank capacity $P$, a small capacity limits sample diversity, whereas a large one increases computational overhead without commensurate performance gains. These results demonstrate that our method is robust within a reasonable range of hyperparameter values and highlight the importance of balancing the frequency and diversity of hard sample replay.

Table 8: Impact of memory bank capacity ($P$) and mining ratio ($\rho$) on BraTS2020 and FLARE2021 performance.

| Mining Ratio ($\rho$) | BraTS2020 | | | | | FLARE2021 | | | | |
|---|---|---|---|---|---|---|---|---|---|---|
| | 0.1 | 0.2 | 0.3 | 0.4 | 0.5 | 0.1 | 0.2 | 0.3 | 0.4 | 0.5 |
| DSC↑ | 0.8520 | 0.8513 | 0.8512 | 0.8507 | 0.8511 | 0.9324 | 0.9311 | 0.9302 | 0.9290 | 0.9281 |

| Bank Capacity ($P$) | BraTS2020 | | | | | FLARE2021 | | | | |
|---|---|---|---|---|---|---|---|---|---|---|
| | 4 | 6 | 8 | 10 | 12 | 4 | 6 | 8 | 10 | 12 |
| DSC↑ | 0.8501 | 0.8512 | 0.8513 | 0.8520 | 0.8515 | 0.9302 | 0.9311 | 0.9316 | 0.9324 | 0.9310 |

