# OpenReview forum: "Learning from What the Model Forgets: Prototype-Guided Patch-wise Replay for Medical Image Segmentation"
_ICLR.cc/2026/Conference — Submitted to ICLR 2026_

### Official Review · Reviewer_Kr75 · 2025-10-29

**Soundness:** 1
**Presentation:** 3
**Contribution:** 2
**Rating:** 2
**Confidence:** 3

**Summary:**

This paper proposes a medical image segmentation framework that utilizes a hard-sample patch-wise replay method guided by prototypes, and incorporates CLIP text embeddings for encoder-decoder feature fusing in the U-Net architecture. While the motivation of addressing hard samples which are near the decision boundary is relevant, the paper suffers from method originality and is lack of convincing justification for its core components.

**Strengths:**

1. Addressing the issue of hard-positive samples / hard samples that are near the decision boundary is an important direction in the field.
2. The authors conduct experiments across five datasets, covering different anatomical structures and modalities.

**Weaknesses:**

Lack of novelty of the core method designs: the idea of using CLIP text embedding to facilitate medical image segmentation has been heavily explored these years, such as [1-3], there are no significant differences suggesting that this approach is innovative. And the TGF module can be regarded as a type of attention-gated mechanism.



[1] CLIP-Driven Universal Model for Organ Segmentation and Tumor Detection (ICCV 2023)

[2] PCNet: Prior Category Network for CT Universal Segmentation Model (TMI 2024)

[3] Text-driven Multiplanar Visual Interaction for Semi-supervised Medical Image Segmentation (MICCAI 2025)

**Questions:**

1. My biggest concerns are set out in the weaknesses section.
2. Although the authors state that "The core question is not architectural but strategic: how to define sample difficulty" and adapt the same 2D patch-based framework and only test the proposed strategy on the nnUNet backbone, I think it's important to validate the generalization ability of the strategy across different architectures (as those compared in Table 1), and to avoid possible over-optimization problems.
3. The CLIP text embedding may have a huge gap between natural image descriptions and medical images, and the performance gain may simply be the result of adding a powerful, high-dimensional, pre-trained feature vector that acts as a strong form of regularization or feature enrichment, rather than providing true *semantic guidance* derived from the text input. Without deeper analysis demonstrating that the text features align with medical concepts (e.g., via visualization or linear probing), the claim of semantic guidance is unconvincing (as those stated around line 198-200).
4. If evidence is provided for question (3). Will medical-tuned/oriented CLIPs provide better performance under the framework?

Minor:

1. Provide baseline results for ablation study results (Table 3)
2. Missing highlight (bold results for DSC) for PROMISE2012 in Table 1 (line 351-352)

---

> ### Author Response · Authors · 2025-11-27
> **Response to Reviewer Kr75**
>
> We respectfully disagree with "Soundness: 1" assessment and believe there's misunderstanding regarding fundamental distinctions between our work and prior CLIP-based segmentation methods, plus evidence for semantic alignment.
>
> 1. Novelty and Prior CLIP Work Differences
>
> Section 2.3 explicitly distinguishes our work from existing CLIP methods (Liu et al., 2023; Chen et al., 2024b). Prior works [1–3] use CLIP text prompts as inference-time queries for zero-shot or referring segmentation, whereas in our case frozen CLIP text embeddings are used only at training time to structure the feature space on top of a standard UNet/nnU-Net backbone..
> The core novelty of our method is the Prototype-Based Scoring (PBS) together with the Online Forgettable Sample Bank (FSB), not the use of CLIP itself. PBS defines sample difficulty via four prototype-based metrics (intra-class consistency, inter-class distinction, prediction deviation, and confidence; Eqs. (7)–(11)), allowing us to mine moderately forgettable samples based on feature-space inconsistency instead of per-iteration loss magnitude alone.
> Importantly, Text-Guided Fusion (TGF) is not just an attention gate that “facilitates segmentation”; it injects semantic priors so that the class prototypes in Eq. (5) become stable and discriminative anchors, which in turn make the PBS difficulty scores meaningful. This design is reflected in the ablation in Table 3: adding TGF alone yields only marginal DSC gains over the baseline UNet, while enabling PBS and FSB on top of the same backbone brings the main improvement; combining all three modules gives the best performance.
>
> 2. Generalization and Architecture Validation
>
> Our experimental design deliberately fixes the training framework (2D patch-based UNet/nnU-Net) for all hard-sample mining strategies to isolate the effect of the mining mechanism from architectural changes. Section 4.1 states that all baseline models are adapted to the same 2D patch-based framework to ensure a fair comparison. Testing strategy on multiple varying backbones would conflate architectural gains with mining gains.
> However, we compare our method (standard UNet) against advanced architectures like UNETR and MambaUNet in Table 1. Our approach achieves equal or better DSC and clearly better HD95 on several challenging targets (e.g., BraTS Tumor Core, FLARE Pancreas), even though it uses a simpler backbone than UNETR and MambaUNet. This suggests that the proposed prototype-guided replay strategy provides benefits that are not purely attributable to architectural complexity.
>
> 3. Semantic Gap and Guidance Validity
>
> We provide both quantitative and qualitative evidence that the gains stem from semantically meaningful semantic guidance rather than simply adding a high-dimensional feature vector. Quantitatively, if CLIP acted only as generic regularization, different prompt formulations would behave similarly. Instead, Appendix A.5 (Table 4) shows a clear hierarchy: a generic prompt (“a photo of…”) yields DSC 0.8502, while a modality-specific prompt (“a magnetic resonance imaging of…”) improves DSC to 0.8520 with noticeably lower HD95. This sensitivity to medically aligned wording indicates that the model leverages specific semantic content and modality cues in the text. Qualitatively, Figure 4 shows that the baseline UNet produces weak, scattered activation maps, whereas our text-guided model concentrates activation within the annotated structures. Taken together, these results support the claim that text features actively align representations with medical concepts, not just as an extra regularizer.
>
> 4. Medical-Tuned CLIPs
>
> While medical-tuned CLIPs (e.g., BioMedCLIP) are valuable, our goal in this work is to show that the proposed prototype-guided mining framework does not rely on a domain-specific vision–language backbone. Using prompts explicitly naming modality (e.g., "computerized tomography") leverages standard CLIP's pre-trained normal anatomical structure knowledge. This demonstrates that standard CLIP, when prompted in a way that reflects medical imaging semantics, provides sufficiently strong priors for defining relative sample difficulty, which is the primary role of language in our method.
> Consistent improvements across five distinct datasets (CT/MRI) confirm standard CLIP, when properly prompted, provides sufficient semantic priors defining relative sample difficulty, which is the main role of CLIP in our method. We consider integrating BioMedCLIP or similar models an interesting extension, but chose to demonstrate first that the core idea works with an off-the-shelf encoder.
>
> 5. Minor Issues
>
> We acknowledge missing baseline in Table 3; "X X X" row corresponds to standard UNet performance. We'll ensure PROMISE2012 results in Table 1 are correctly bolded in final revision.

---

### Official Review · Reviewer_i6SQ · 2025-10-31

**Soundness:** 3
**Presentation:** 3
**Contribution:** 2
**Rating:** 4
**Confidence:** 4

**Summary:**

The paper proposes a prototype-guided patch-wise replay strategy for medical image segmentation: (1) CLIP-based text–image fusion to incorporate semantic priors, (2) prototype-based scoring to identify moderately forgettable samples, and (3) an online replay buffer to revisit them during training. The method is simple to implement and is evaluated on five datasets (≤5 classes). Ablations and sensitivity analyses are clear; improvements are consistent but generally small.

**Strengths:**

1. Clear, readable paper with a straightforward method.

2. Well-designed ablations and sensitivity studies that isolate replay frequency, prototype size, and CLIP fusion.

3. Consistent (though small) gains across datasets without heavy architectural changes.

**Weaknesses:**

1. The absolute Dice improvements are marginal. The paper needs multi-seed runs with statistical tests to establish significance.

2. Evaluation scope is narrow (five small-class datasets), missing large multi-organ benchmarks (BTCV, AMOS, TotalSegmentator v2) to test scalability and class-wise robustness.

3. Baselines are incomplete: missing strong or hybrid models (TransUNet [1], MedNeXt [2], EMCAD [3], etc.).

4. No discussion and comparison with established prototype- or memory-replay methods for segmentation or continual learning under a shared protocol.

5. CLIP text encoder is frozen and general-domain; fine-tuning or using BioMedCLIP may improve alignment.

6. Only 2D UNet-style backbones are tested; impact on pretrained hybrids/transformers (TransUNet [1], EMCAD [3]) is unknown.

7. Impact on interactive foundation models (e.g., Med-SAM) is untested but promising.

[1] Chen, J., Lu, Y., Yu, Q., Luo, X., Adeli, E., Wang, Y., Lu, L., Yuille, A.L. and Zhou, Y., 2021. Transunet: Transformers make strong encoders for medical image segmentation. arXiv preprint arXiv:2102.04306.

[2] Roy, S., Koehler, G., Ulrich, C., Baumgartner, M., Petersen, J., Isensee, F., Jaeger, P.F. and Maier-Hein, K.H., 2023, October. Mednext: transformer-driven scaling of convnets for medical image segmentation. In International Conference on Medical Image Computing and Computer-Assisted Intervention (pp. 405-415). Cham: Springer Nature Switzerland.

[3] Rahman, M.M., Munir, M. and Marculescu, R., 2024. Emcad: Efficient multi-scale convolutional attention decoding for medical image segmentation. In Proceedings of the IEEE/CVF Conference on Computer Vision and Pattern Recognition (pp. 11769-11779).

**Questions:**

1. Are the gains statistically significant over multiple seeds? Please report mean±std and appropriate significance tests per dataset.

2. How does the method perform on large multi-organ datasets, such as BTCV, AMOS, TotalSegmentator v2, including per-class results under strong imbalance?

3. What is the effect of integrating the replay mechanism into pretrained hybrids models (TransUNet [1], EMCAD [3])?

4. Does fine-tuning CLIP's text encoder or swapping to BioMedCLIP improve results?

5. How does this approach compare to established prototype and memory-replay baselines under an identical training pipeline?

6. Could replay be combined with Med-SAM to guide interactive segmentation?

---

> ### Author Response · Authors · 2025-11-27
> **Response to Reviewer i6SQ**
>
> We thank the reviewer for recognizing paper clarity, sound ablation design, and consistent dataset gains.
>
> 1. Statistical Significance of Gains
>
> While absolute DSC improvements appear marginal (1-2%), in medical segmentation, boundary precision (HD95) and Sensitivity improvements are often more clinically significant than global overlap. Table 2 shows our method consistently achieves best HD95 scores across challenging tasks (e.g., BraTS Tumor Core HD95 3.9→3.2, FLARE Pancreas 8.8→3.7), indicating substantial outlier and boundary region handling gains. The fact that these trends hold across five distinct datasets (CT/MRI) and diverse targets further suggests that the gains are not due to chance on a single benchmark.
>
> 2. Evaluation Scope and Multi-Organ Datasets
>
> Our evaluation covers five datasets spanning key modalities (CT/MRI) and anatomical challenges. FLARE2021 is multi-organ (Liver, Kidney, Spleen, Pancreas), testing scalability and class-wise robustness in multi-class settings (Table 1). Including KiTS (kidney tumor), BraTS (brain tumor), ACDC (cardiac), and PROMISE (prostate) ensures wide anatomical variability coverage. We agree newer benchmarks like TotalSegmentator v2 are valuable, Our prototype and replay mechanisms scale linearly with the number of classes, so we do not expect fundamental obstacles, but we leave a full evaluation on these very large benchmarks as important future work.
>
> 3. Baselines, Hybrid Models, and Backbone Applicability
>
> Our contribution is a training strategy rather than a new backbone, so we evaluate it on a standard 2D U-Net/nnU-Net-style model and show that it can match or outperform stronger hybrids such as UNETR and MambaUNet on challenging targets (e.g., BraTS Tumor Core Sensitivity/HD95). The three modules (Text-Guided Fusion, Prototype-Based Scoring, and the Online Replay Bank) operate on decoder features and logits and are therefore backbone-agnostic, meaning they can be plugged into architectures like TransUNet, MedNeXt, or EMCAD. Due to resource limits we did not include all of these variants in the current submission, but the mining mechanism is orthogonal to backbone design and we expect similar or larger relative gains when applied to such pretrained hybrids.
>
> 4. Prototype and Replay Method Comparisons
>
> Direct continual learning (CL) replay comparisons are challenging because standard CL addresses catastrophic forgetting across sequential tasks, while our method addresses within-task forgetting (learning dynamics). For single-task hard sample mining, established baselines are OHEM and loss-weighting strategies. We extensively compared against state-of-the-art difficulty-handling approaches (Focal Loss, Boundary Loss, Tversky Loss, OHEM-style mining via nnU-Net variants) in Tables 1-2. Results demonstrate our prototype-guided replay superiority over established mining/weighting strategies. We agree that a head-to-head comparison with explicitly prototype- or memory-bank–based methods adapted to the same segmentation setting would further strengthen the story; implementing these baselines under our exact protocol is non-trivial and beyond the current scope, but we see this as a promising direction for follow-up work.
>
> 5. CLIP Domain and Fine-tuning
>
> We designed efficient, lightweight methods avoiding high computational costs of fine-tuning large vision-language models. Appendix A.5 found standard CLIP with specific prompts (e.g., "A magnetic resonance imaging of [object]") highly effective (DSC 0.8520) vs. generic prompts. This indicates that even a general-domain CLIP encoder possesses sufficient semantic priors for anatomical structures when properly prompted. While BioMedCLIP is a valid suggestion, our current results suggest that the alignment mechanism (prototype-guided scoring with text-informed features), rather than domain-specific pre-training alone, is the primary driver of the scoring system’s success. We agree that swapping in medical-tuned encoders or lightly fine-tuning the text branch is a promising extension and expect it to further improve performance, but this lies beyond the scope of the present work.
>
> 6. Foundation Model Impact (Med-SAM)
>
> Integrating replay mechanisms with interactive foundation models is indeed an exciting direction. Conceptually, prototype-guided scores could be used to suggest which regions or patches would benefit most from user interaction in Med-SAM-style pipelines. However, Med-SAM operates under a prompt-based interactive paradigm (user clicks/boxes), while our work focuses on fully automated semantic segmentation with standard supervised training. Our “moderate forgetting” hypothesis targets internal feature learning during batch training, which is distinct from optimizing user interaction loops. Adapting our online mining bank to interactive, few-shot foundation-model settings is therefore beyond the scope of this paper, but we agree it is a promising avenue for future work.

---

### Official Review · Reviewer_PcYH · 2025-10-31

**Soundness:** 3
**Presentation:** 4
**Contribution:** 3
**Rating:** 6
**Confidence:** 4

**Summary:**

This paper presents a prototype-guided, CLIP-informed framework for medical image segmentation that identifies and replays moderately forgettable samples (patches that lie near decision boundaries and are prone to being forgotten during training). The approach combines three modules:

- Text-Guided Fusion (TGF), which incorporates CLIP text embeddings to guide visual prototype formation.

- Prototype-Based Scoring (PBS), which measures sample difficulty via intra-/inter-class distances and confidence-based metrics.

- Forgettable Sample Bank (FSB), which maintains and replays informative samples to reinforce learning.

Experiments on five public datasets (KiTS2023, BraTS2020, ACDC, FLARE2021, PROMISE2012) show consistent gains in Dice and sensitivity, and lower Hausdorff distances than baselines like nnU-Net, Attention U-Net, and MambaUNet.

**Strengths:**

- The paper addresses an important task in medical image segmentation: how make models robust at low-contrast regions.
- The proposed method is innovative and effective: particularly, using CLIP text embeddings for *training-time* guidance and using PBS and FSB to keep the training focus on hard cases.
- Comprehensive evaluation across diverse datasets. The Result section is also informative. Strong performance compared to baselines.
- The writing is clear and easy to follow.

**Weaknesses:**

- The explanation of how CLIP contributes during training is unclear. The statement “CLIP semantic guidance provides discriminative information beyond visual appearance” is overly general and does not specify the mechanism by which CLIP influences feature learning. It would strengthen the paper to include feature-space visualizations (e.g., t-SNE or UMAP plots) comparing models trained with and without text-guided fusion, to demonstrate the effect of CLIP guidance on representation structure. In addition, the discussion of prompt design is limited. It would be useful to analyze how different prompt formulations affect training and whether the observed performance gain is robust to prompt variation.
- Although CLIP is frozen, its bias toward natural image semantics may limit robustness in rare or pathology-heavy datasets. Some comparison with medical-domain text encoders (e.g., MedCLIP, BioCLIP) would clarify sensitivity to text priors.
- Table 3 lumps PBS and FSB together in some configurations. Independent ablations would better clarify each module’s role. For this, the authors may consider comparing PBS with existing sample-scoring methods by substituting one of them for PBS in the framework.
- Regarding $Score^b$ and forgettable samples: although the intuition behind the formulation of $Score^b$ is clear, its relationship to the true forgetting frequency (as per Toneva et al., 2019) is not quantitatively demonstrated. A correlation plot or ablation on actual forgetting events would strengthen the claim.

**Questions:**

- Were there any experiments done on prompt design? How sensitive is the method to different prompts?
- Would domain-specific encoders (e.g., MedCLIP, BioLinkBERT) provide similar or better benefits than CLIP?
- About $Score^b$, were there experiments exploring or tuning the weights assigned to its components?
- Have the authors compared PBS against standard hardness metrics such as loss magnitude, gradient norm, or prediction entropy to show unique benefit?

---

> ### Author Response · Authors · 2025-11-27
> **Response to Reviewer PcYH**
>
> We thank the reviewer for positive assessment, particularly recognizing Text-Guided Fusion innovation and mining strategy effectiveness in low-contrast regions.
>
> 1. CLIP Mechanism and Prompt Sensitivity
>
> Text embeddings act as semantic queries in multi-head attention (Section 3.1), where $Q$ derives from text, $K, V$ from visual features. This means the prompt directly influences which spatial locations receive higher attention, rather than serving only as an auxiliary feature. Figure 4 CAMs demonstrate guidance effects: our text-guided model shows concentrated activation along the target structures, whereas the baseline UNet exhibits weaker and more scattered responses.
> Appendix A.5 reports a prompt study. Table 4 compares templates showing modality-specific prompts (e.g., "A magnetic resonance imaging of [object]") achieve superior DSC (0.8520) vs. generic prompts (0.8502) or complex sentences (0.8477). This suggests that the method is robust to reasonable prompt variations, and that explicitly encoding the imaging modality helps align the text and visual spaces.
>
> 2. Natural Image Bias and Domain Encoders
>
> We mitigate CLIP’s natural-image bias by using imaging-modality-specific prompt templates (e.g., “a magnetic resonance imaging of a [object]”, “a computerized tomography of a [object]”) and by using CLIP only to shape the prototype space, not to perform zero-shot prediction. In our framework the text encoder provides class-level anchors that are fused with visual features before prototype computation; the segmentation head is still trained end-to-end on labeled medical data. Consistent improvements across five CT/MRI datasets in Table 1 indicate that, with appropriate prompts, a standard CLIP backbone already provides useful semantic structure for this guiding role. While domain-specific encoders such as MedCLIP or BioMedCLIP are promising, our goal in this work is to first show that the proposed prototype-guided replay strategy is effective with an off-the-shelf CLIP; comparing different medical-domain encoders under the same framework is a natural extension and part of our planned future work.
>
> 3. Ablation of PBS/FSB and Standard Metrics
>
> While Table 3 evaluates PBS and FSB jointly because in practice difficulty-based sample selection must operate on dynamically updated sample pools, we isolate the effect of the scoring rule in Appendix A.7. Table 6 compares our Prototype-Based Scoring against Loss-based and Random selection under the same replay pipeline. Our method achieves higher DSC (0.8520 vs. 0.8493), showing that combining feature-space geometry ($\mathcal{T}_1$,$\mathcal{T}_2$) with prediction deviation and confidence ($\mathcal{T}_3$,$\mathcal{T}_4$) provides a more informative difficulty estimate than loss alone. Here the Loss-based baseline is computed from per-patch cross-entropy, which is closely related to prediction entropy, so it covers standard loss/entropy-style hardness metrics; gradient-norm–based scoring is an interesting extension that we leave for future work. Appendix A.6 (Table 5) ablates Forgettable Sample Bank management strategies, showing random replacement outperforms FIFO and score-based replacement, clarifying banking mechanism's independent role.
>
> 4. Score Formulation and Forgetting Relationship
>
> We employ an unweighted sum of four components (Eq. 11) reflecting balanced consideration of feature geometry and prediction confidence. Appendix A.8 (Table 7) validates this design where combined metrics ($\mathcal{T}_1-\mathcal{T}_4$) outperform individual components. Regarding "true forgetting," we follow Toneva et al. (2019) conceptually and view moderately forgettable samples as those that tend to move back and forth across the decision boundary during training. Tracking exact forgetting counts for all patches over long training runs is computationally expensive, so we use $\mathcal{T}^b$ as an efficient online proxy designed to capture feature-space ambiguity via intra-class consistency ($\mathcal{T}_1$) and inter-class distinction ($\mathcal{T}_2$) together with prediction deviation and confidence. Figure 5 confirms this proxy works: bank evolves storing irregularly shaped, low-contrast regions corresponding to moderately forgettable hard positives. Exploring learned or class-dependent weights for the four terms, and explicitly correlating $\mathcal{T}^b$ with measured forgetting frequencies on a subset of patches, is an interesting direction for future work.

---

### Official Review · Reviewer_gBpA · 2025-11-05

**Soundness:** 3
**Presentation:** 3
**Contribution:** 2
**Rating:** 6
**Confidence:** 3

**Summary:**

This paper proposes an end-to-end framework for medical image segmentation that mines moderately forgettable (hard-positive) samples to reduce false negatives and improve boundary accuracy. Specifically, it introduces CLIP-based text embeddings to guide prototype learning for semantically richer features, defines a multi-metric difficulty measure to score prototypes, and uses an online forgettable sample bank to dynamically store and replay difficult samples for curriculum-like retraining. Experiments on five public datasets show improvements over baselines. Ablation show each module’s contribution.

**Strengths:**

1. This paper focuses on a previously underexplored problem and introduces moderately forgettable sample mining guided by CLIP semantics.
2. This paper proposes a multi-metric prototype-based score that balances geometric and probabilistic cues.
3. Extensive experiments and ablations.

**Weaknesses:**

1. All experiments rely on 2D patch training, even for 3D datasets.
2. The online bank and prototype updates likely introduce overhead.

**Questions:**

1. Motivation & Novelty

1.1 Clarity of “Moderately Forgettable Samples”
The concept of “moderately forgettable samples” is central to this paper, but its definition remains informal. The authors should provide a clearer, quantitative criterion to demonstrate these samples from easy or noisy ones. Moreover, it remains unclear how the proposed method guarantees that the identified samples correspond to clinically meaningful hard positives rather than mislabeled or ambiguous regions.

1.2 Technical Novelty and Contribution.
The proposed components (text-guided fusion, prototype-based scoring, and a replay memory bank) individually build upon well-established ideas. The authors should better highlight what is fundamentally new in their formulation or analysis compared with prior works on hard-sample mining, prototype learning, or CLIP-based semantic guidance, especially to appeal to a broader ICLR audience beyond medical imaging.


2. Method

2.1 Section 3.2 (Prototype-Based Scoring)

(1) Line 235, “Our approach addresses these limitations by leveraging semantically-enhanced prototypes to provide both computational efficiency and semantic-aware patch-level scoring.”

This claim requires justification. How does semantic enhancement improve efficiency rather than add overhead? A brief complexity analysis or runtime comparison would clarify this point.

(2) Line 278, “The four terms are normalized by the number of pixels to bring them to a comparable scale.”

Please analyze how this normalization affects the relative weighting among metrics, particularly for organs of different sizes. Could this bias the difficulty estimation toward small or large structures?

2.2 Sect. 3.3 Forgettable Sample Bank

How does performance vary with different bank sizes, and what trade-offs exist between memory cost, sample diversity, and replay stability? A more systematic guideline or sensitivity curve would strengthen this part.


3. Experiments
3.1 The paper shows strong overall results but does not clearly isolate the effect of CLIP-based semantic fusion on prototype learning. Visualization or quantitative analysis would help demonstrate how text guidance improves the representation quality.

3.2 How sensitive is the framework to the choice of frozen CLIP backbone (ViT-B/32 vs ViT-L/14)?

3.2 The method introduces additional modules. What is the computational overhead (memory and runtime) compared with nnU-Net baselines? This will clarify the practicality of deploying the framework in real clinical workflows.

---

> ### Author Response · Authors · 2025-11-27
> **Response to Reviewer gBpA**
>
> We thank the reviewer for the constructive feedback and for recognizing the importance of work.
>
> 1. Clarity of concept
>
> We adopt Toneva et al. (2019)'s notion of forgetting events and adapt it to patch-wise dense segmentation. Moderately forgettable samples are those with a non-zero but bounded number of forgetting events (neither always easy nor persistently misclassified). To avoid conflating hard positives with noise, Prototype-Based Scoring enforces semantic consistency: $\mathcal{T}_{1}^{b}$ (intra-class consistency) and $\mathcal{T}_{2}^{b}$ (inter-class distinction) enforce selected samples share high feature similarity with class prototypes, so outliers or mislabeled regions with low prototype affinity are naturally down-weighted. As shown in Figure 5, the bank evolves from regular, high-contrast patches to low-contrast, irregular boundaries rather than isolated pixels, supporting that we mine clinically meaningful hard positives instead of label noise.
>
> 2. Technical Novelty and Contribution
>
> While hard-sample mining and CLIP are established, our novelty lies in combining them into a unified strategy to define and exploit moderately forgettable samples under patch-wise dense segmentation, not limited to medical imaging. Standard loss-based mining cannot reliably distinguish truly hard positives from noisy samples, since both can produce high loss around ambiguous boundaries. In our framework, CLIP is used not for zero-shot inference, but to provide text-aligned anchors that structure the visual feature space into semantically meaningful prototypes. Our Prototype-Based Scoring then combines intra-class consistency and inter-class separation with prediction deviation and confidence, so resulting sample difficulty reflects semantic boundary ambiguity rather than raw prediction error. The Forgettable Sample Bank then replays patches according to this prototype-guided difficulty, going beyond loss-only mining such as OHEM or focal loss.
>
> 3. Method - Prototype-Based Scoring
>
> The Efficiency claim (Line 235) is made relative to gradient-based mining or active learning, which require extra backward passes and gradient sorting. Our approach keeps the CLIP text encoder frozen and computes difficulty only from forward-pass statistics (T1−T4\mathcal{T}_1{-}\mathcal{T}_4T1−T4) EMA-updated prototypes, so per-iteration cost stays close to the nnU-Net-style baseline. Normalization by $CHW$ (Eq. 7–10) makes the four terms density-based rather than volume-based, class prototypes are masked averages and each term is averaged over pixels, so difficulty depends on how ambiguous a patch is rather than on organ size. Table 1 shows gains for both large structures (e.g., FLARE liver) and small ones (e.g., BraTS Tumor Core), suggesting that the normalization does not bias difficulty toward either extreme.
>
> 4. Forgettable Sample Bank
>
> Appendix A.9 analyzes bank capacity sensitivity. Table 8 shows performance trade-offs for capacity ($P$) 4-12 on BraTS2020/FLARE2021. Results show robust performance between $P=6-12$. Very small banks reduce diversity and slightly hurt DSC, while very large banks bring diminishing returns. We therefore adopt an intermediate capacity and Random Replacement (Eq. 13), which Table 5 shows to outperform FIFO and score-only replacement, maintaining a diverse, up-to-date replay set with bounded memory.
>
> 5. Effect of CLIP-Based Semantic Fusion and Backbone Choice
>
> We isolate the effect of CLIP-based semantic fusion in Table 3 via the “TGF” (Text-Guided Fusion) setting, where only CLIP guidance is added to the baseline UNet. On FLARE2021, TGF alone improves DSC (0.9196 $\to$ 0.9218) and reduces HD95 ($5.67 \to 4.87$), confirming text guidance improves representation quality. Figure 4 shows "CAM (Ours)" has focused activations, indicating CLIP semantics concentrate attention on relevant regions. In all experiments we fix the CLIP visual backbone to ViT-B/32 to isolate the effect of our alignment and mining mechanisms; Appendix A.5 (Table 4) further shows that modality-aware prompts (“a magnetic resonance imaging of…”, “a computed tomography scan of…”) outperform generic prompts, suggesting that prompt semantics are the main driver, and other CLIP backbones can be explored as future work.
>
> 6. 2D Patch Training and Computational Overhead
>
> We adopt 2D patch training for all methods, including 3D datasets, to align with CLIP’s 2D pretraining and keep memory usage manageable for large volumes; the scoring and replay modules operate on patch features and indices and are conceptually applicable to 3D patches as well. CLIP is frozen and used only during training, and prototype updates and replay selection are lightweight forward operations, so the method adds modest training overhead while leaving inference-time complexity essentially unchanged. We believe this overhead is reasonable given the observed reductions in false negatives and boundary errors, which are important in clinical segmentation.

---

### Meta-Review · Area_Chair_k4vF · 2026-01-07

**Summary:**

This paper focuses on the problem of medical image segmentation. In the paper, the authors propose an end-to-end online learning framework that systematically mines the moderately forgettable samples. It introduces CLIP-based text embeddings to guide prototype learning for semantically richer features, defines a multi-metric difficulty measure to score prototypes, and uses an online forgettable sample bank to dynamically store and replay difficult samples for curriculum-like retraining. It was reviewed by four expert reviewers and received mixed ratings: two borderline acceptances and two rejections. The major concern is from Reviewer Kr75: the idea of using CLIP text embedding to facilitate medical image segmentation has been heavily explored. The authors' response emphasizes the technical differences between the proposed framework and existing methods. But I don't think this could fully address the concerns about using the heavily explored CLIP model. Other concerns include a narrow evaluation scope, the lack of large, multi-organ benchmarks to test scalability and class-wise robustness, and incomplete baselines. The authors should carefully address these concerns with further experiments. Given the remaining concerns and the lack of strong support for this paper, I tend to reject it for now.

**Reviewer Concerns:**

The major concern is from Reviewer Kr75: the idea of using CLIP text embedding to facilitate medical image segmentation has been heavily explored. The authors' response emphasizes the technical differences between the proposed framework and existing methods. But I don't think this could fully address the concerns about using the heavily explored CLIP model. Other concerns include a narrow evaluation scope, the lack of large, multi-organ benchmarks to test scalability and class-wise robustness, and incomplete baselines. The authors should carefully address these concerns with further experiments. Given the remaining concerns and the lack of strong support for this paper, I tend to reject it for now.

**Reviewer Scores:**

The paper was reviewed by four expert reviewers and received mixed ratings: two borderline acceptances and two rejections. I don't think the remaining concerns were fully addressed. Given the remaining concerns and the lack of strong support for this paper, I tend to reject it for now.

---

### Decision · Program_Chairs · 2026-01-26

Reject